# Complete Chloroplast Genome Sequences of Five *Ormosia* Species: Molecular Structure, Comparative Analysis, and Phylogenetic Analysis

**Jianmin Tang** [1,2,3] **, Rong Zou** [1] **, Xiao Wei** [1,*] **and Dianpeng Li** [1]

[1] Guangxi Key Laboratory of Plant Functional Substances and Resources Sustainable Utilization, Guangxi Institute of Botany, Guilin 541006, China; tjm@gxib.cn (J.T.); zr@gxib.cn (R.Z.); ldp@gxib.cn (D.L.)

[2] Faculty of Agriculture, Saga University, Saga 840-8502, Japan

[3] The United Agricultural Research Department, Graduate School, Kagoshima University, Kagoshima 890-0065, Japan

[*] Correspondence: wx@gxib.cn

**Abstract:** The genus *Ormosia* comprises approximately 130 species, which are found in tropical regions all over the world. The taxonomy and evolutionary history are still ambiguous due to the lack of a solid phylogeny at the species level. Due to the fast advancement of high-throughput sequencing technology, it is now possible to retrieve the full chloroplast (cp) genome sequence, providing a molecular basis for phylogenetic analysis. Five species of *Ormosia* were used in this work, and their whole cp genomes were sequenced. One circular, quadripartite-structured molecule, ranging from 169,797 to 173,946 base pairs in length, was present in all five *Ormosia* species. The cp genomes of the five newly sequenced *Ormosia* species were further compared with the published cp genomes of *O. pinnata*. Ten hypervariable regions (Pi > 0.025) were uncovered in the *Ormosia* chloroplast genomes, among which *petN-psbM* showed the highest Pi value. Phylogenetic analysis revealed that *O. microphylla* and *O. olivacea* clustered with strong support. *O. balansae* was resolved as a sister to *O. pinnata*, and they were further found to be sisters to the remaining 13 *Ormosia* species. The cp genes of *O. elliptica* showed a close relationship to *O. hosiei*, and *O. semicastrata* clustered with *O. emarginata*. Taken together, the comprehensive analysis of the complete cp genomes of five *Ormosia* species offers valuable insight and information for reconstructing their phylogeny and sheds light on the evolutionary dynamics of the chloroplast genome in *Ormosia*.

**Keywords:** *Ormosia*; chloroplast genome; phylogenetic analysis; SSR; IR variations

## 1. Introduction

*Ormosia* is a genus of flowering plants belonging to the Fabaceae family, which is one of the largest families of flowering plants. *Ormosia* species are usually trees or big bushes found in tropical and subtropical regions across the world, primarily in Asia, Africa, and the Americas [1]. *Ormosia* species are characterized by their pinnate leaves, which consist of several leaflets arranged on either side of a central stem, and their showy, pea-like flowers that range in color from red to yellow. The fruits of *Ormosia* species are generally legumes, which contain extremely hard seeds that are often brightly colored in shades of red, orange, brown, and black [2]. The seeds have little nutritional value and contain poisonous alkaloids [3], but they are commonly utilized in jewelry and handcrafts [4]. In addition to the spectacular seeds, *Ormosia* is unique for its arborescent habit, and certain species are highly prized for their sturdy and handsome wood, which is used in furniture, flooring, and construction [5].

There are around 130 species in the genus *Ormosia* [6], and phylogenetic analysis of the genus dates back over a century. Prain (1990, 1904) divided it into two groups, each with four subsections, based on pod structure and seed characteristics [7,8]. Merrill and

Chen (1943) then conducted a survey of Chinese and Indochinese *Ormosia* species, which they classified into 15 series based mostly on fruit and seed characteristics [9]. Even though both categories were based on morphology, their categorization differed from Prain's in numerous ways, suggesting that morphological parameters alone are insufficient for correct classification. In a recent study by Torke et al., a comprehensive molecular phylogeny of *Ormosia* was constructed using nuclear and plastid DNA sequences, including *matK* and *trnL-F*, from a diverse range of 82 species [10]. The varied species found in southern China and Southeast Asia were classified into two clades. However, *matK* and *trnL* intron sequences were found to differ slightly between species. Efforts should be made to improve the phylogenetic resolution and identify the sources of conflicting categorization.

Next-generation sequencing methods hold a lot of promise in this field because, compared to Sanger sequencing, they provide much more data and make statistically rigorous comparisons across DNA regions and genomes more easily [11]. In recent years, the study of plant evolutionary biology has benefited greatly from the utilization of chloroplast (cp) genomes. These genomes have emerged as valuable tools due to their genetic stability, unique genome structure, and relatively faster evolutionary rate compared to mitochondria [12]. The recent use of Illumina sequencing to obtain complete cp genomes of five Chinese *Ormosia* species was a significant first step in this field [13]. However, just a few genomic resources of this genus have been investigated [14–18]. At present, there are only around ten sequences of a complete cp genome of *Ormosia* species in GenBank.

In this work, the cp genomes of five *Ormosia* species were sequenced: *O. microphylla*, *O. semicastrata*, *O. olivacea*, *O. balansae*, and *O. elliptica*. All of these species are planted in southern China and are valuable wood species for housing construction, furniture, wheels, and pipes. Additionally, there is still uncertainty in the categorization of these five species. We carried out a cp genome-based investigation and provide a detailed description of the cp genome assembly, annotations, and simple sequence repeats (SSRs). We conducted phylogenetic analyses of *Ormosia* based on the whole cp genome sequences, using newly sequenced species together with previously published ones. This study aimed to achieve three main objectives: (i) To characterize the structure of the *Ormosia* cp genome, providing insight into its organization and arrangement. (ii) To identify highly divergent regions within the *Ormosia* cp genome that could serve as potential DNA barcodes, which would be useful for species identification and molecular characterization of *Ormosia* species. (iii) To investigate the evolutionary relationships among different *Ormosia* species using cp genome sequences, contributing to the understanding of the genetic divergence and phylogenetic patterns within the genus *Ormosia*.

## 2. Materials and Methods

### 2.1. Plant Materials and DNA Extraction

Fresh leaves of one plant each of five *Ormosia* species—*O. microphylla*, *O. semicastrata*, *O. olivacea*, *O. balansae*, and *O. elliptica*—were collected from the Guilin Botanical Garden, Guangxi, China (25°4′14.88″ N, 110°17′57″ E). Fresh leaves (>1.0 g) were used for the extraction of total genomic DNA. The Magnetic Plant Genomic DNA Kit (TIANGEN Biotech, Beijing, China) was employed following the manufacturer's instructions. The DNA quality was evaluated using both electrophoresis in a 1% agarose gel and a TBS-380 Mini-Fluorometer (Invitrogen, Waltham, MA, USA).

### 2.2. Chloroplast Genome Sequencing and Assembly

Initially, 1 µg of DNA was utilized for library construction. The DNA sample underwent sonication to fragment into 300–500 bp fragments. Subsequently, the fragmented DNA was subjected to end-polishing, A-tailing, and ligation with full-length adaptors for sequencing. Polymerase chain reaction (PCR) amplification was carried out using the cBot TruSeq PE Cluster Kit v3-cBot-HS (Illumina, San Diego, CA, USA). Following PCR, the products were purified using the AMPure XP system (Beckman Coulter Inc., Brea, CA, USA). To generate sequencing libraries, the VAHTS Universal Plus DNA Library Prep Kit

for Illumina (Vazyme, Nanjing, China) was employed, adhering to the manufacturer's instructions. Each sample's sequences were assigned unique index codes. The size distribution of the resulting libraries was evaluated using the Agilent 2100 Bioanalyzer, and quantification was performed using real-time PCR. For cluster generation, the index-coded samples were clustered using the cBot Cluster Generation System (Illumina) as per the manufacturer's instructions. The libraries were subsequently subjected to paired-end sequencing with 150 bp reads using the Illumina NovaSeq 6000 platform.

The quality of the raw paired-end reads was assessed using FastQC v0.11.7 software. Following the quality evaluation, the data were integrated into optimum contigs by de novo assemblers (Fast-Plast, https://github.com/mrmckain/Fast-Plast, accessed on 27 November 2022, and GetOrganelle, https://github.com/Kinggerm/GetOrganelle, accessed on 27 November 2022). The cp sequences of *Ormosia hosiei* (MG813874), *O. formosana* (MT258921), *Zollernia splendens* (MN709880), *Lespedeza maritima* (MG867570), and *Laguncularia racemose* (MH551145) were downloaded from GenBank and used as the seed sequence for *O. balansae*, *O. semicastrata*, *O. elliptica*, *O. olivacea*, and *O. microphylla*, respectively.

After that, the obtained cp genomes were annotated using the PGA software (https://github.com/quxiaojian/PGA, accessed on 5 December 2022) and Geseq software (https://chlorobox.mpimpgolm.mpg.de/geseq.html, accessed on 5 December 2022) with default settings. Manual corrections were made to ensure accuracy. The resulting gene map was visualized using the online OGDraw v1.2 software [19]. Five complete cp genomes were deposited at GenBank (accession nos.: OQ862759 (*Ormosia balansae*), OQ862760 (*O. semicastrata*), (OQ862761 (*O. elliptica*), (OQ862762 (*O. olivacea*), and (OQ862763 (*O. microphylla*)). Relative synonymous codon usage (RSCU) and amino acid frequency in the protein-coding gene region were determined by MEGA-X [20].

*2.3. Repeat Sequences and SSRs*

For the analysis of repeat sequences and simple sequence repeats (SSRs), the cp genome sequence of O. *pinnata* (NC_064393.1) obtained from GenBank and the five newly sequenced cp genomes of *Ormosia* species were utilized. A Perl script called MISA was employed to identify SSRs in the complete cp genome sequences of the six *Ormosia* species. The thresholds used for different SSR lengths were as follows: mononucleotides (10 repeats), dinucleotides (6 repeats), trinucleotides (5 repeats), tetranucleotides (5 repeats), pentanucleotides (5 repeats), and hexanucleotides (5 repeats).

To detect repeat sequences, the REPuter program [21] was employed, which identified four types of repeats: palindromic, forward, reverse, and complement repeats. The following criteria were used to identify repeat sequences within the cp genome: (1) a Hamming distance of 3, (2) a minimum size of 30 base pairs, and (3) a sequence identity of at least 90%.

*2.4. Variations and Divergent Hotspot Regions of cp Genomes*

To create the sequence variation map, the mVISTA comparative genomics server [22] was utilized, with the annotation of the *O. balansae* cp genome serving as the reference. The variations in inverted repeat (IR) sequences, including expansion and contraction events, were assessed using the online IRscope program (https://irscope.shinyapps.io/irapp/, accessed on 20 December 2022). To identify the hotspots of intergeneric divergence, a sliding window analysis was conducted using DnaSP v5.10 software [23]. A window length of 600 base pairs (bp) was selected, and the step size was set to 200 bp.

*2.5. Phylogenomic Reconstruction Based on cp Genomes*

The phylogenomic analysis was conducted using the maximum likelihood method based on a dataset consisting of the five newly sequenced cp genomes of *Ormosia* species and ten *Ormosia* species downloaded from GenBank. *Sophora velutina* and *S. tomentosa* were defined as outgroups. To align the sequences, the Multiple Sequence Alignment Based on Fast Fourier Transform (MAFFT) program [24] was employed, and conserved

sequences were screened out by Gblocks v0.91b software [25]. The best substitution model, GTR + G, was selected in the jModelTest v2.1.7 program [26], and the maximum likelihood method was applied in MEGA 6.0 software to infer the phylogenetic relationships using 1000 bootstrap replicates [27]. Maximum likelihood (ML) was performed using RAxML v8.2.12 software. Samples were collected at every 1000 generations. The resulting phylogenetic tree was visualized using FigTree v1.4.4 (http://tree.bio.ed.ac.uk/software/figtree/, accessed on 15 January 2023).

## 3. Results

### 3.1. Overall cp Genome Features of Five Ormosia Species

The cp genomes of the five *Ormosia* species exhibited a circular structure (Figure 1) with a quadripartite organization. Their lengths ranged from 169,797 to 173,946 base pairs, with *Ormosia microphylla* having the largest genome and *O. elliptica* the smallest. The cp genomes consisted of a large single copy (LSC) region (70,849–73,937 bp) and a small single copy (SSC) region (18,272–18,785 bp), separated by a pair of inverted repeats (IRa and IRb, with lengths of 80,676–81,224 bp) (Table 1). The total GC content was 35.71–36.16%, suggesting almost equal values among the five complete *Ormosia* cp genomes. Additionally, the GC content distribution varied among different regions, with the LSC, SSC, and IR regions displaying GC contents of 33.45–33.98%, 29.73–30.24%, and 39.16–39.42%, respectively (Table 1). The RSCU values of each codon of five *Ormosia* cp genomes are shown in Table S1. Totally, 29 preferentially used codons were found in the five *Ormosia* cp genomes, with RSCU values that were relatively comparable across species.

**Table 1.** Statistics on basic features of chloroplast genomes of five *Ormosia* species.

| | | *Ormosia balansae* | *Ormosia microphylla* | *Ormosia elliptica* | *Ormosia olivacea* | *Ormosia semicastrata* |
|---|---|---|---|---|---|---|
| Total | Length (bp) | 170,836 | 172,973 | 169,797 | 172,829 | 173,946 |
| | GC% | 35.92 | 35.83 | 36.16 | 35.84 | 35.71 |
| LSC | Length (bp) | 71,096 | 73,163 | 70,849 | 73,055 | 73,937 |
| | GC% | 33.79 | 33.62 | 33.98 | 33.65 | 33.45 |
| | Length (%) | 41.62 | 42.30 | 41.73 | 42.27 | 42.51 |
| IR | Length (bp) | 81,038 | 81,178 | 80,676 | 81,154 | 81,224 |
| | GC% | 39.21 | 39.18 | 39.42 | 39.18 | 39.16 |
| | Length (%) | 47.44 | 46.93 | 47.51 | 46.96 | 46.70 |
| SSC | Length (bp) | 18,702 | 18,632 | 18,272 | 18,620 | 18,785 |
| | GC% | 29.74 | 29.86 | 30.24 | 29.87 | 29.73 |
| | Length (%) | 10.95 | 10.77 | 10.76 | 10.77 | 10.80 |
| Protein-coding | Length (bp) | 81,768 | 81,777 | 81,445 | 80,838 | 76,590 |
| | GC% | 37.42 | 37.46 | 37.53 | 37.37 | 37.29 |
| | Length (%) | 47.86 | 47.28 | 47.97 | 46.77 | 44.03 |
| tRNA | Length (bp) | 2793 | 2737 | 2792 | 2795 | 2804 |
| | GC% | 53.35 | 53.49 | 53.51 | 53.42 | 53.35 |
| | Length (%) | 1.63 | 1.58 | 1.64 | 1.62 | 1.61 |
| rRNA | Length (bp) | 9062 | 9062 | 9062 | 9056 | 9054 |
| | GC% | 55.40 | 55.44 | 55.44 | 55.43 | 55.47 |
| | Length (%) | 5.30 | 5.24 | 5.34 | 5.24 | 5.21 |

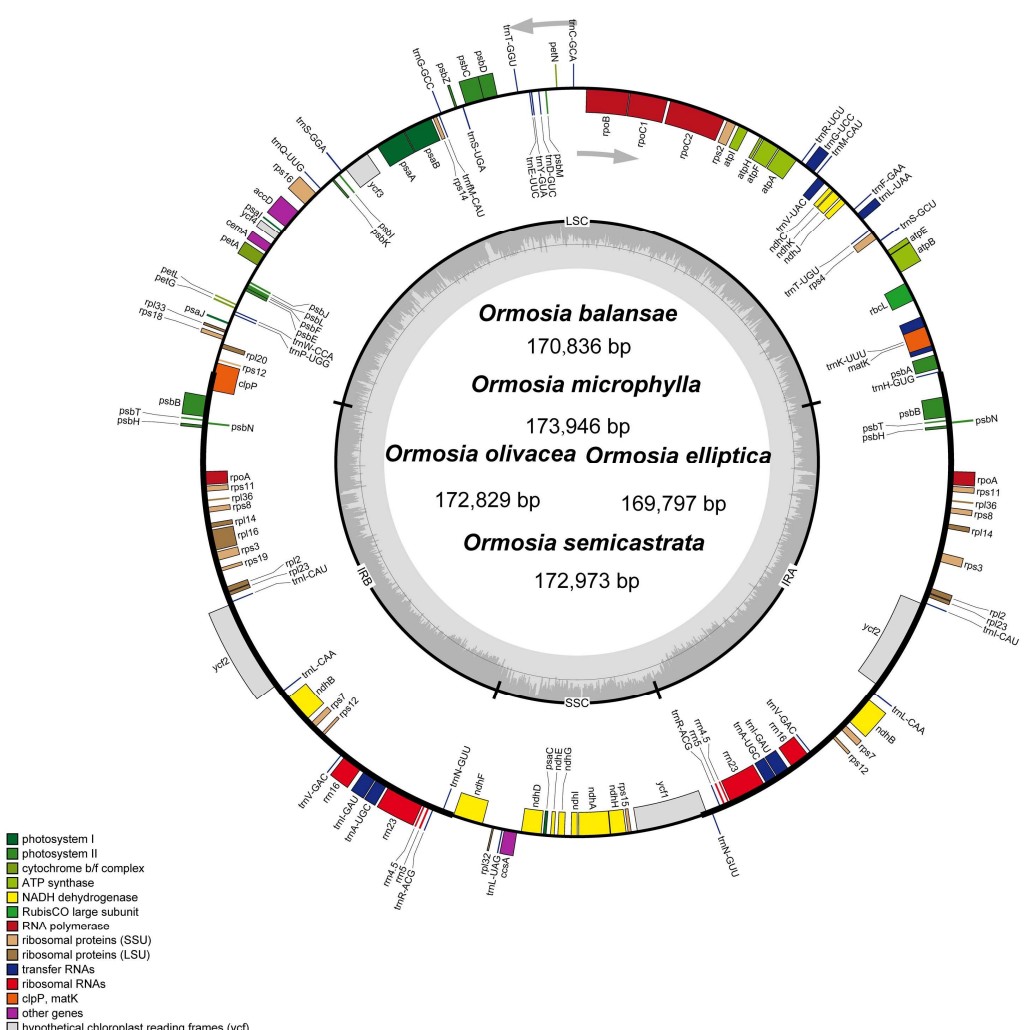

**Figure 1.** Gene map of chloroplast genomes of five *Ormosia* species. Directed by arrow, genes located on the outer circle indicate clockwise transcription, while those on the inner circle indicate counterclockwise transcription. Functional groups of genes are visually distinguished through color coding. The inner circle exhibits varying shades of gray, with darker shades representing the GC content of the chloroplast genome and lighter shades representing the AT content.

### 3.2. IR Expansion and Contraction

The IR/LSC and IR/SSC junction regions of five newly sequenced cp genomes of *Ormosia* species and the cp genome of *O. pinnata* downloaded from GenBank were compared to assess potential expansion or contraction events. While the length of IR regions in the six *Ormosia* species remained similar, minor shifts in the IR/SC borders were observed for *Ormosia* species (Figure 2). In *O. olivacea*, *O. elliptica*, and *O. balansae*, the *ycf1* pseudogene spanned the IRb/SSC boundary, extending 1–32 bp into the SSC region, whereas it is found 1 bp away from the border in *O. pinnata* and *O. microphylla*. The IRb/LSC junction is found in the *clpP* region in all six species, and the IRa/SSC junction is located in the *ycf1* region. The length of the IRs differs between species: 2022–2084 bp for *clpP* and 5300–5348 bp for normal *ycf1* gene. Additionally, a noncoding sequence of 3–9 bp was found between the IRa/LSC boundary and the 3'-end of the *trnH-GUG* gene in the LSC region.

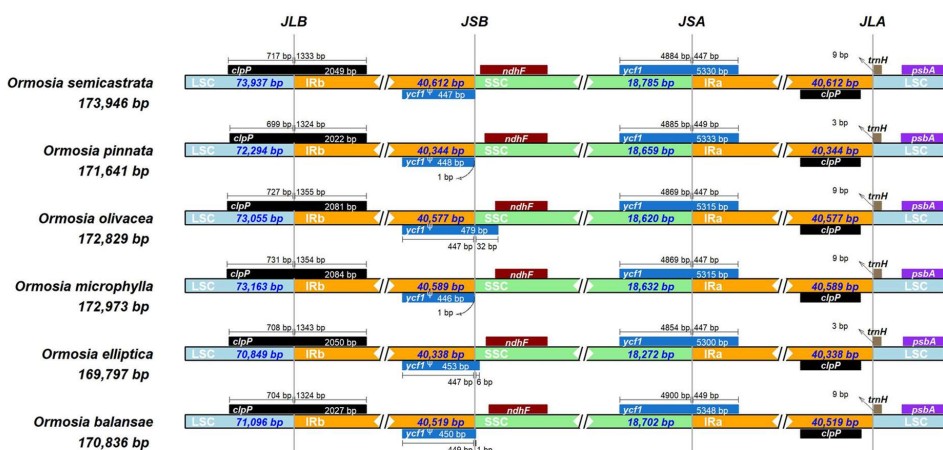

**Figure 2.** Comparison of LSC, SSC, and IR regional boundaries of chloroplast genomes between six *Ormosia* species. JLB, junction line between LSC and IRb; JSB, junction line between IRb and SSC; JSA, junction line between SSC and IRa; and JLA, junction line between IRa and LSC. The start and end of each gene from the junctions has been shown with arrows. Ψ, pseudogene. The cp genome sequence of *O. pinnata* (NC_064393.1) was obtained from GenBank.

### 3.3. Variations and Divergence Hotspot Regions of cp Genomes

The intergeneric divergence of cp genome sequences was assessed by calculating the percentage of identity among six *Ormosia* species, using *O. balansae* as the reference genome (Figure 3). Overall, the arrangement of the cp genome in *Ormosia* exhibited a high level of collinearity, with a conserved gene order. There was a considerable degree of similarity across the six species, with noncoding region variation being substantially larger than coding area variation.

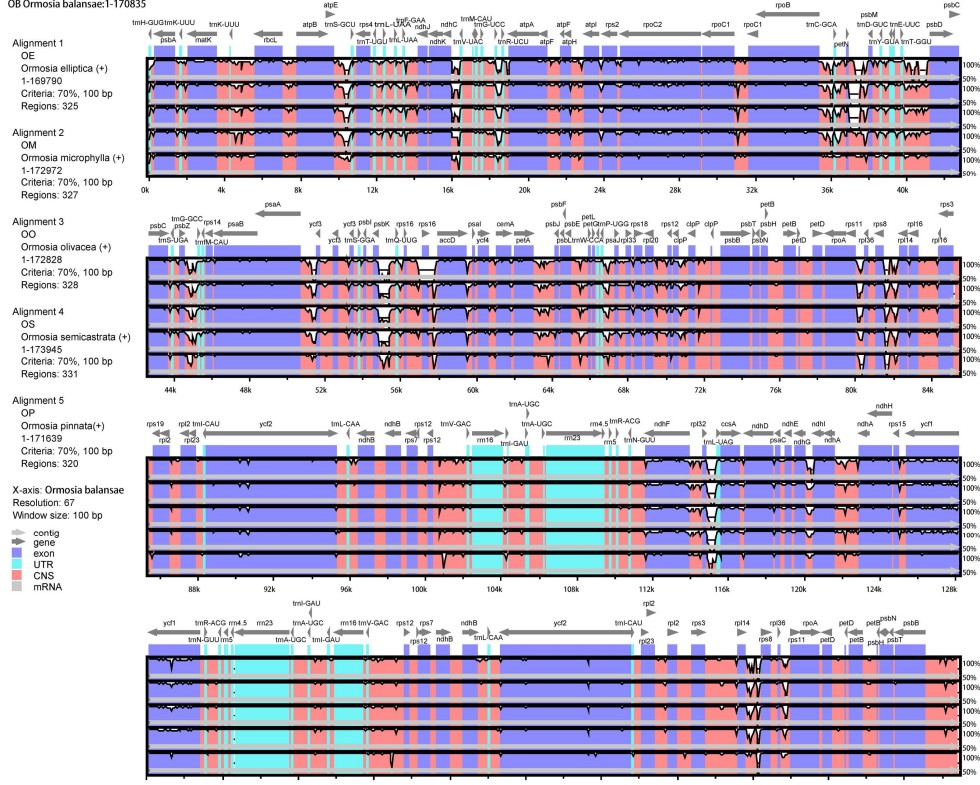

**Figure 3.** Comparison of chloroplast genomes via annotation of *Ormosia balansae* as a reference. The vertical scales represent the percentage of identity, ranging from 50% to 100%, while the horizontal

scale represents the coordinates within the chloroplast genome. Gray arrows indicate genes with their orientation. The different regions of the genome, including exons, introns, untranslated regions (UTRs), and conserved non-coding sequences (CNSs), are color-coded for easy visualization. The cp genome sequence of *O. pinnata* (NC_064393.1) was downloaded from GenBank.

The nucleotide diversity (pi) values within 600 bp segments were analyzed to identify divergence hotspots (Figure 4). The pi values varied from 0 to 0.075, with higher genetic diversity observed in the LSC (average pi = 0.013) and SSC (average pi = 0.011) areas than in the IR area (average pi = 0.006), indicating that the IR region was more conserved than the LSC and SSC regions across the six cp genomes. The highest variable region was *petN−psbM*, with a Pi value of 0.073, while seven additional hypervariable regions (Pi > 0.025) were identified in the *Ormosia* cp genomes, as shown in Figure 4. Nine hypervariable regions are located in the LSC or SSC region, while there is only one hypervariable region (*ycf2*) in the IR region.

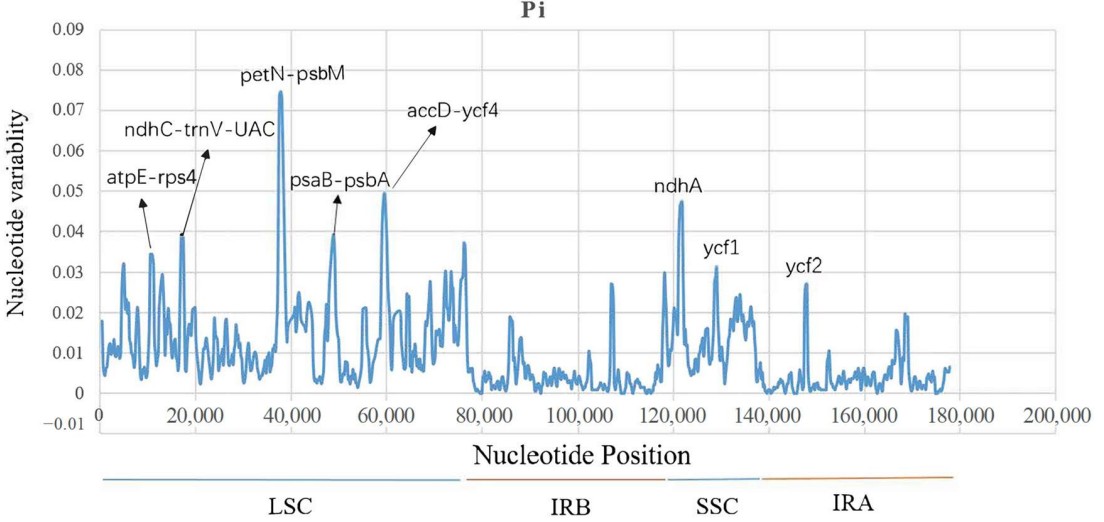

**Figure 4.** Sliding window analysis of six *Ormosia* chloroplast genomes (window length: 600 bp; step size: 200 bp).

*3.4. Repeat Analysis*

SSRs and large repeats in the cp genomes of six *Ormosia* species were investigated. It was observed that the *Ormosia* cp genomes contained a range of 79 to 102 SSRs (Figure 5a). Among these SSRs, the majority were located in the LSC regions (70.0–77.0%), with the number varying within species, ranging from 46 in *O. balansae* to 70 in *O. microphylla*. Four types of SSRs (mono-, di-, tri-, and tetra-nucleotide repeats) were observed in the cp genomes of the six *Ormosia* species (Figure 5b). Mono-nucleotide repeats were found to be the most prevalent, accounting for 69.62 and 77.01% of total SSRs in *O. balansae* and *O. elliptica*, respectively. Tri-nucleotide repeats were detected in *O. microphylla*, *O. olivacea*, and *O. semicastrata*, while tetra-nucleotide repeats were only detected in *O. balansae* and *O. pinnata*.

The investigation of large repeats in the cp genomes of the six *Ormosia* species revealed the presence of 40 to 43 large repeats. Among the species, *O. semicastrata* and *O. olivacea* did not exhibit any complementary repeats in their cp genomes (Figure 5c). Except for *O. elliptica*, which had more palindromic repeats than forward repeats, the number of forward repeats was larger in all investigated species. The length distribution of repeats is illustrated in Figure 5d. Although the number of repeats was similar among the six *Ormosia*, the length of repeats varied. Repeats with a length of 2130 bp were predominant in the cp genomes of *O. elliptica*, *O. microphylla*, *O. olivacea*, and *O. semicastrata*, accounting for 88.1 to 91.5% of total repeats, whereas repeats with a length of 3140 bp only accounted for 1.6 to

7.5%. The proportion of repeats with a length of 3140 bp was much higher in *O. balansae* and *O. pinnata*, making up 29.9 and 36.2% of total repeats, respectively.

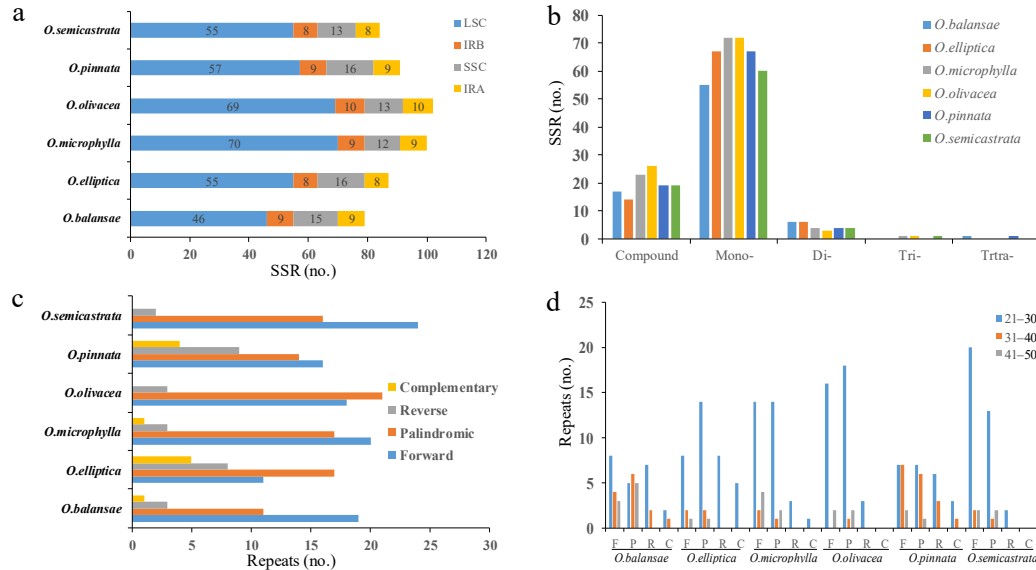

**Figure 5.** Analysis of simple sequence repeats (SSRs) and repeated sequences in chloroplast genomes of *Ormosia* species: (**a**) the number of SSRs; (**b**) the type and frequency of each identified SSR; (**c**) the identification of four types of repeats; and (**d**) the frequency of repeat occurrence based on length. The cp genome sequence of *O. pinnata* (NC_064393.1) was obtained from GenBank.

### 3.5. Phylogenomic Analysis

The cp genomic protein-coding regions of 15 species of *Ormosia* were used to construct a phylogenetic tree using the maximum likelihood method, with *Sophora tomentosa* and *S. velutina* defined as outgroups (Figure 6). In terms of the five newly sequenced cp genomes, *O. microphylla* and *O. olivacea* clustered with strong support (100 BS). *O. balansae* was resolved as a sister to *O. pinnata* (100 BS), and both were further found to be sisters to the remaining 13 *Ormosia* species. The cp genome of *O. elliptica* exhibited a close relationship with *O. hosiei* (100 BS), and *O. semicastrata* clustered with *O. emarginata* (100 BS).

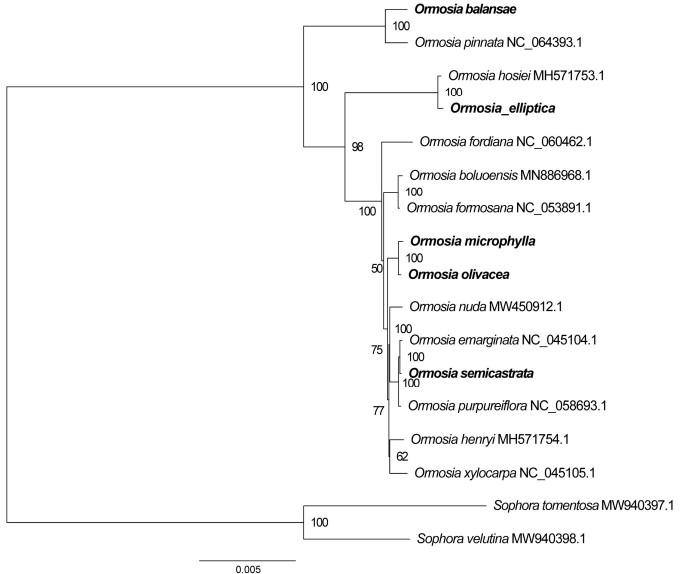

**Figure 6.** Phylogenetic tree constructed using cp genomic protein-coding regions of five *Ormosia* species and their related species, obtained from GenBank.

## 4. Discussion

The analysis of cp genome sequences provides comprehensive information for phylogenetic analysis of plants, and DNA barcoding using cp markers enables reliable identification of plant species [28]. In this study, we present the cp genomes of five *Ormosia* species and compare them to another species, *O. pinnata*, to gain insight into the differences among the cp genomes described above. Furthermore, we conducted phylogenetic analysis by incorporating the complete cp genomes of these species along with nine other *Ormosia* species. By studying the cp genome sequences of these species, we aim to enhance our understanding of the evolutionary dynamics within the *Ormosia* genus.

The structure of the complete cp genomes of all five studied *Ormosia* species is comparable to that of the majority of plants, consisting of a single circular quadripartite molecule [29–31]. The cp genomes of the five *Ormosia* species examined in this study were relatively larger than those of most angiosperms [32]. For instance, the cp genomes of the *Dalbergia* species, belonging to the Fabaceae family, have been reported to be approximately 156,000 in size [33]. The longer sequences of *Ormosia* cp genomes are ascribed to the IR regions, which are twice as long as those of *Dalbergia* species. The size of the five studied *Ormosia* cp genomes is comparable to that of previously reported *Ormosia* species, ranging from 170,811 to 174,128 bp [13]. The cp genomes of most land plants are highly conserved, and the *chlB*, *chlL*, *chlN*, and *trnP-GGG* are observed to be missing in flowering plants [34]. The deletion of the above four genes was found in the cp genomes of all five *Ormosia* species. GC content plays a crucial role in sequence stability, with higher GC content indicating increased DNA density and a more conserved and rigid sequence [35]. Our findings indicate a high degree of conservation in the cp genomes of *Ormosia* species, with similar GC content observed in each region (LSC, SSC, and IR) among the species (Table 1). Variation in GC content was observed among different regions, with the IR regions exhibiting higher GC content, primarily due to the presence of rRNAs [29]. The high GC content in these regions plays a critical role in maintaining the base composition of cp genomes and stabilizing their overall structure.

The SSRs present in the cp genomes of plants have implications for a variety of fields, including genetics, conservation biology, phylogenetics, and plant breeding. They serve as useful genetic markers for studying the genetic diversity of plant populations, understanding the evolutionary relationships among plant species, and reconstructing their phylogenetic history, and can be utilized in plant breeding programs to develop improved crop varieties by identifying SSR markers linked to desirable traits [36]. Analysis of the cp genomes of six *Ormosia* species revealed the presence of five types of SSRs, ranging from mononucleotide to compound repeats. Among these, mononucleotide repeats were the most abundant, consistent with the findings reported by Liu et al. [13] for five species within the same genus. Our study identified a total of 543 SSRs in the cp genomes of the six *Ormosia* species, indicating a wealth of SSR polymorphism information in these genomes. These findings have implications for the identification and analysis of genetic diversity among *Ormosia* species. Large repeats were critical for studying genome reorganization, rearrangement, and phylogeny, as well as causing substitutions and insertions in the cp genome [37]. We detected 40–43 large repeats in five *Ormosia* species, but the types and numbers of each type differed between species. The complementary repeats were missing in *O. semicastrata* and *O. olivacea* cp genomes, which may also play a role in the genetic diversity and evolution of different *Ormosia* species.

The expansion and contraction of IR region borders are the primary causes of cp genome size variation and are important in species evolution [38]. In our study, we observed both similarities and differences in the junctions of LSC, SSC, and IR among the six species under comparison. Additionally, certain genes, including *ycf1*, *clpP*, and *trnH*, exhibited shifts at the borders (Figure 2). Previous reports have indicated that the varying lengths of cp genomes may be created by expansion and contraction occurring in the IR regions [31]. By contrast, we observed that the variation in *Ormosia* cp genome length was

attributed to the differences in length of non-coding sequences in the LSC region, which was consistent with Liu's report [13].

Determining variations in cp genomes is critical to comprehending the evolution and genomic structure of chloroplasts [39]. In this study, it was observed that the IR regions exhibited lower variability compared to the LSC and SSC regions, which aligns with the findings of a previous study [30]. Since non-coding regions generally undergo faster mutation rates than coding regions [40], the highly variable regions in *Ormosia* cp genomes are mostly found in intergenic spacers, such as *atpE-trnS*, *ndhC-trnV*, *petN-trnD*, *psbK-trnQ*, and *rpl32-trnL*. The rRNA genes showed no significant variation in coding regions between the six *Ormosia* species (Figure 3), which was consistent with prior findings that rRNA gene sequences were highly conserved [41]. Natural selection has little effect on nucleotide substitutions in intergenic spacer and intron regions, as well as pseudogenes that are not translated into proteins [42]. As a result, a non-coding area can be used to deduce the evolutionary history [43]. Certain coding areas with relatively substantial sequence variation have also been found to be a useful source for interspecies phylogenetic research [10]. Several divergence hotspot regions (e.g., *aspE-rps4*, *petN-psbM*, *psaB-psbA*, *accD-ycf4*, *ndhA*, *ycf1*, and *ycf2*) were found by calculating and comparing the nucleotide diversity (Pi) values (Figure 4). These regions exhibit significant variability and can potentially serve as valuable molecular markers in future phylogenetic studies [44].

The importance of cp genomes in reconstructing phylogenetic relationships and understanding evolutionary history has been demonstrated [45]. Cp genome markers such as *matK*, *trnL-F*, *ndhF*, *trnH-psbA*, *rpoB*, and *ycf* have been widely used in taxonomy and DNA barcoding investigations [46,47]. Torke et al. conducted a Bayesian analysis of nuclear and plastid marker sequences and classified *O. elliptica*, *O. olivacea*, and *O. microphylla* into *Ormosia* clade I, and *O. semicastrata*, *O. balansae*, and *O. pinnata* into clade II [10]. Here, we confirmed a close relationship between *O. balansae* and *O. pinnata*, and between *O. olivacea* and *O. microphylla*. Both *O. balansae* and *O. pinnata*, increasingly cultivated as ornamental street trees in southern China, are attractive for their thick leaves and red seeds. *O. olivacea* and *O. microphylla*, which grow on hillsides at an altitude over 600 m, have great economical value for their wood. Interestingly, our study found that *O. elliptica* was sister to *O. henryi*, which had been placed in its own series by Merrill and Chen (1943) due to its relatively long hilum [9]. The morphological characteristics of *O. elliptica* are extremely close to those of *O. henryi*, except for the shape of the leaf and pod. Most nodes were allocated with high BP values in our study, except *O. henryi* and *O. xylocarpa*. Liu et al. also reported a low BP value for five *Ormosia* species [13]. Therefore, more information on the cp genomes of *Ormosia* species is needed to enhance the resolution of the phylogeny.

**Supplementary Materials:** The following supporting information can be downloaded at: https://www.mdpi.com/article/10.3390/horticulturae9070796/s1, Table S1: Codon usage in five *Ormosia* species chloroplast genome.

**Author Contributions:** Conceptualization, X.W.; methodology, R.Z.; validation, X.W.; formal analysis, J.T. and R.Z.; investigation, X.W. (sampling), D.L. (sequencing) and J.T. (data analysis); writing—original draft preparation, J.T.; writing—review and editing, D.L.; visualization, X.W. and D.L.; supervision, X.W.; funding acquisition, X.W. All authors have read and agreed to the published version of the manuscript.

**Funding:** This study was supported by the projects of the National Key Research and Development Program (No. 2022YFF1300700), the Chinese Academy of Sciences 'Light of West China' Program (2022), the Guangxi Forestry Science and Technology Promotion Demonstration Project (2023LYKJ03 and [2022]GT23), the Guangxi Key Laboratory of Plant Functional Phytochemicals Research and Sustainable Utilization Independent Project (No. ZRJJ2022-2), and the Guilin Innovation Platform and Talent Plan (20210102-3).

**Institutional Review Board Statement:** Not applicable.

**Informed Consent Statement:** Not applicable.

**Data Availability Statement:** All data cited in the study are publicly available.

**Conflicts of Interest:** The authors declare no conflict of interest.

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
