# Peer review of "Complete Chloroplast Genome Sequences of Five Ormosia Species: Molecular Structure, Comparative Analysis, and Phylogenetic Analysis"

_horticulturae, doi:10.3390/horticulturae9070796_

Round 1

Reviewer 1 Report

Need more information/justification on why did you choose these 5 species? Is it just because they are planted and available in Guilin Botanical Garden- Southern China? or any other reasons?

Author Response

Need more information/justification on why did you choose these 5 species? Is it just because they are planted and available in Guilin Botanical Garden- Southern China? or any other reasons?

Response: Thank you for your inquiry regarding the selection of the five species for our study. There are three main reasons: The first reason is that these five species are extensively cultivated in the Guangxi region of China. This makes them readily available for research and provides us with a suitable sample pool to study their characteristics and properties. The second reason is the fact that their tree trunks are composed of precious wood. Lastly, these five species have been chosen due to the ambiguity in their classification. We have made corresponding revisions in the revised version.

This part  (Lines 331-342) is suggested under the conclusion.

Response:  Upon revisiting the suggestion, we would like to clarify that the content in question pertains to the discussion of phylogenetic relationships, including comparisons with other relevant literature. As such, it is better suited for the discussion section, where it aligns with the logical flow of the manuscript and allows for a comprehensive exploration of the topic.

Reviewer 2 Report

The article horticulturae-2464943 provided a comprehensive understanding of the chloroplast genome of five Ormosia species. Thus, the article horticulturae-2464943 is slightly suited to the Horticulturae as per aims and scope. However, to improve the quality of the manuscript, the Reviewer has some minor comments.

 - Comment 1: The methods section is slightly acceptable. However, could the Authors provide some parts related to the codon usage bias analyses. Also, please cite the references for the software and web-based tools mentioned in this section.

  - Comment 2: The structural variation, codon usage, oligonucleotide repeats and polymorphic loci are very interesting parts in this manuscript. If possible, please improve these parts in the Discussion section.

 - Comment 3: The references did not follow exactly the style of Horticulturae. Please, improve this part.

 - Comment 4: Please check for grammatical and spelling mistakes. Many words should be italicized. Some duplicated words should be noted to check. All abbreviations should be carefully checked in the whole text.

Author Response

- Comment 1: The methods section is slightly acceptable. However, could the Authors provide some parts related to the codon usage bias analyses. Also, please cite the references for the software and web-based tools mentioned in this section.

Response: thanks a lot for your suggestion. We have added codon usage analysis in the modified version. In addition, references or URLs were added to the software listed in this section.

  - Comment 2: The structural variation, codon usage, oligonucleotide repeats and polymorphic loci are very interesting parts in this manuscript. If possible, please improve these parts in the Discussion section.

Response: thanks for your suggestion. We have improved discussion section in the revised manuscript. We believe that the additions will enhance the manuscript

 - Comment 3: The references did not follow exactly the style of Horticulturae. Please, improve this part.

Response: We appreciate your attention to detail and for bringing this matter to our attention. In the revised manuscript, we have made the necessary adjustments to match the referencing style of Horticulturae accurately.

Comments on the Quality of English Language

 - Comment 4: Please check for grammatical and spelling mistakes. Many words should be italicized. Some duplicated words should be noted to check. All abbreviations should be carefully checked in the whole text.

Response: Thank you for your valuable feedback regarding grammatical and spelling mistakes in our manuscript. In the revised manuscript, we have conducted a comprehensive proofreading to ensure the accuracy of grammar and spelling.

Reviewer 3 Report

The structure of the MS is quite typical for articles on sequencing and analysis of chloroplast genomes.

A few notes are given below.

Line 45: Prain (1990, 1904) – add reference number

Lines 52, 74-80, 195, 221: Ormosia and O. balansae in Italic

Lines 63, 221,261,266, 267, 285, 286,292, 296, 325, 326: chloroplast > cp

Line 75: chloroplast (cp) genome > cp genome

Line 129: REPuter – add URL or reference

Line 148:Gblock v0.91b – add URL or reference

Lines 152-154 should be deleted. It concerns the MrBayes program!

Lines 405-407 and Ref.22 should be deleted. Please clarify MEGA or RaXML was used!

Line 155: FigTree– add URL or reference

Line 135: mVISTA  – add URL or reference

Line 136: O. balansae in Italic

Line 201: delete “in Figure 3”; scale > scales

Line 208: “a 600 bp segment” > “600 bp segments”

Line 223: delete “Figure 5a

Section 3.4: Figures 5b and 5c are not mentioned

Section 3.5: “protein-coding regions” (line 250) or “complete chloroplast genomes” (line 257) were used for tree construction?

Author Response

Line 45: Prain (1990, 1904) – add reference number

Response: thanks for your suggestion. We have added the reference number in the revised version.

Lines 52, 74-80, 195, 221: Ormosia and O. balansae in Italic

Response: sorry for the carelessness. We have revised the words accordingly.

Lines 63, 221,261,266, 267, 285, 286,292, 296, 325, 326: chloroplast > cp

Response: we have used abbreviation as you suggested.

Line 75: chloroplast (cp) genome > cp genome

Response: we have used abbreviation in the revised version.

Line 129: REPuter – add URL or reference

Response: thanks for your suggestion. We have added the reference in the revised version.

Line 148: Gblock v0.91b – add URL or reference

Response: We have added the reference in the revised version.

Lines 152-154 should be deleted. It concerns the MrBayes program!

Response: thanks for your suggestion. We have deleted the sentence.

Lines 405-407 and Ref.22 should be deleted. Please clarify MEGA or RaXML was used!

Response: we have deleted the reference. Thanks for your suggestion.

Line 155: FigTree– add URL or reference

Response: we have added URL in the revised version.

Line 135: mVISTA  – add URL or reference

Response: We have added the reference in the revised version.

Line 136: O. balansae in Italic

Response: sorry for the carelessness. We have revised the words accordingly.

Line 201: delete “in Figure 3”; scale > scales

Response: we have modified the sentence.

Line 208: “a 600 bp segment” > “600 bp segments”

Response: we have modified the sentence.

Line 223: delete “Figure 5a”

Response: we have deleted ‘Figure 5a’ in the revised version.

Section 3.4: Figures 5b and 5c are not mentioned

Response: we have added ‘Figures 5b and 5c’ in the appropriate places.

Section 3.5: “protein-coding regions” (line 250) or “complete chloroplast genomes” (line 257) were used for tree construction?

Response: sorry for the carelessness. We used protein-coding regions of cp genomes for phylogenetic tree construction. We have revised the sentence in Line 257.

Reviewer 4 Report

An article „Complete Chloroplast Genome Sequences of Five Ormosia Species: Molecular Structure, Comparative and Phylogenetic Analysis” is well written. The section ‘Materials and Methods” is adequate to conduced bioinformatic experiments and contains all necessary descriptions but there is one mistake in lines 119-120 related to accession numbers. OQ862760 (O. microphylla) in NCBI states for O. semicastrata and opposite - OQ862763 states for O. semicastrata.

The genus names should be in Italic, example: line 136, 195, 221 and others.

In line 163, abbreviation IRs brings to confusion and should be deleted..

In Figure 2 there are two ycf1 genes described – one about 400bp around JSB and another of about 5000bp around JSA. In NCBI the firs one is properly described as pseudogene in all accessions. This should be corrected in the Figure 2 as well as thorough the text

Author Response

An article „Complete Chloroplast Genome Sequences of Five Ormosia Species: Molecular Structure, Comparative and Phylogenetic Analysis” is well written. The section ‘Materials and Methods” is adequate to conduced bioinformatic experiments and contains all necessary descriptions but there is one mistake in lines 119-120 related to accession numbers. OQ862760 (O. microphylla) in NCBI states for O. semicastrata and opposite - OQ862763 states for O. semicastrata.

Response: sorry for the carelessness. We have corrected it in the revised version.

The genus names should be in Italic, example: line 136, 195, 221 and others.

Response: thanks for the suggestion. We have carefully reviewed the text and consistently applied the appropriate formatting to enhance the clarity and accuracy of the species names mentioned throughout the document.

In line 163, abbreviation IRs brings to confusion and should be deleted.

Response: we have deleted ‘IRs’ as you suggested.

In Figure 2 there are two ycf1 genes described – one about 400bp around JSB and another of about 5000bp around JSA. In NCBI the first one is properly described as pseudogene in all accessions. This should be corrected in the Figure 2 as well as thorough the text.

Response: we have corrected the first as ycf1 pseudogene and marked in Figure 2.